# STABILIZED MEDICAL IMAGE ATTACKS

**Gege Qi**[1], **Lijun Gong**[1*], **Yibing Song**[2*], **Kai Ma**[1], **Yefeng Zheng**[1]
[1] Tencent Jarvis Lab, Shenzhen, China
[2] Tencent AI Lab, Shenzhen, China
`lijungong@tencent.com  yibingsong.cv@gmail.com  yefengzheng@tencent.com`

## ABSTRACT

Convolutional Neural Networks (CNNs) have advanced existing medical systems for automatic disease diagnosis. However, a threat to these systems arises that adversarial attacks make CNNs vulnerable. Inaccurate diagnosis results make a negative influence on human healthcare. There is a need to investigate potential adversarial attacks to robustify deep medical diagnosis systems. On the other side, there are several modalities of medical images (e.g., CT, fundus, and endoscopic image) of which each type is significantly different from others. It is more challenging to generate adversarial perturbations for different types of medical images. In this paper, we propose an image-based medical adversarial attack method to consistently produce adversarial perturbations on medical images. The objective function of our method consists of a loss deviation term and a loss stabilization term. The loss deviation term increases the divergence between the CNN prediction of an adversarial example and its ground truth label. Meanwhile, the loss stabilization term ensures similar CNN predictions of this example and its smoothed input. From the perspective of the whole iterations for perturbation generation, the proposed loss stabilization term exhaustively searches the perturbation space to smooth the single spot for local optimum escape. We further analyze the KL-divergence of the proposed loss function and find that the loss stabilization term makes the perturbations updated towards a fixed objective spot while deviating from the ground truth. This stabilization ensures the proposed medical attack effective for different types of medical images while producing perturbations in small variance. Experiments on several medical image analysis benchmarks including the recent COVID-19 dataset show the stability of the proposed method.

## 1 INTRODUCTION

Computer Aided Diagnosis (CADx) has been widely applied in the medical screening process. The automatic diagnosis benefits doctors to efficiently obtain health status to avoid disease exacerbation. Recently, Convolutional Neural Networks (CNNs) have been utilized in CADx to improve the diagnosis accuracy. The discriminative representations improve the performance of medical image analysis including lesion localization, segmentation and disease classification. However, recent advances in adversarial examples have revealed that the deployed CADx systems are usually fragile to adversarial attacks (Finlayson et al., 2019), e.g., small perturbations applied to the input images can deceive CNNs to have opposite conclusions. As mentioned in Ma et al. (2020), the vast amount of money in the healthcare economy may attract attackers to commit insurance fraud or false claims of medical reimbursement by manipulating medical reports. Moreover, image noise is a common issue during the data collection process and sometimes these noise perturbations could implicitly form adversarial attacks. For example, particle contamination of optical lens in dermoscopy and endoscopy and metal/respiratory artifacts of CT scans frequently deteriorate the quality of collected images. Therefore, there is a growing interest to investigate how medical diagnosis systems respond to adversarial attacks and what we can do to improve the robustness of the deployed systems.

While recent studies of adversarial attacks mainly focus on natural images, the research of adversarial attacks in the medical image domain is desired as there are significant differences between

---

*L.Gong and Y. Song are corresponding authors. The code is available at `https://github.com/imogenqi/SMA`

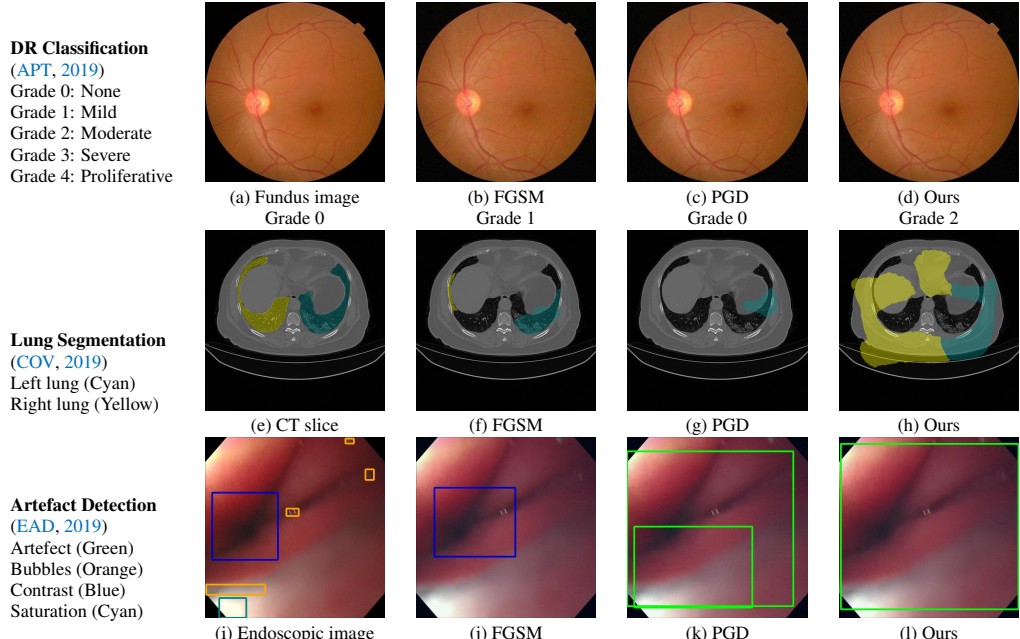

**DR Classification**
(APT, 2019)
Grade 0: None
Grade 1: Mild
Grade 2: Moderate
Grade 3: Severe
Grade 4: Proliferative

(a) Fundus image
Grade 0

(b) FGSM
Grade 1

(c) PGD
Grade 0

(d) Ours
Grade 2

**Lung Segmentation**
(COV, 2019)
Left lung (Cyan)
Right lung (Yellow)

(e) CT slice

(f) FGSM

(g) PGD

(h) Ours

**Artefact Detection**
(EAD, 2019)
Artefect (Green)
Bubbles (Orange)
Contrast (Blue)
Saturation (Cyan)

(i) Endoscopic image

(j) FGSM

(k) PGD

(l) Ours

Figure 1: Adversarial attacks on medical images. A clean fundus image is shown in (a) and correctly classified as "None" during diabetic retinopathy grading. The perturbations from FGSM (Goodfellow et al., 2014) attack successfully (i.e., grading as "Mild") in (b) while PGD (Madry et al., 2017) fails (i.e., grading still as "None"). A clean CT slice is shown in (e) where the lung is correctly segmented. The perturbations from FGSM do not attack completely (i.e., cyan mask is still accurate) in (f) while PGD works in (g). A clean endoscopic image detection result is shown in (i). FGSM and PGD are not effective to fail the detector completely. The perturbations produced by SMIA decrease the analysis performance across different medical image datasets as shown in (d), (h) and (l).

two domains. Beyond regular RGB cameras, there are various types of medical imaging equipments (e.g., Computed Tomography (CT) scanners, ultrasound transducers and fundus cameras) to generate dramatically different images. Fig. 1 shows three examples where an image captured from fundus camera is in (a), an image captured from the CT scanner is in (e) and an endoscopic video frame is in (i). As can be seen in the figure that these three images have little in common. The huge data variance across different modalities of medical images brings more challenges to develop a technology that works for all the modalities. In addition, existing investigations on medical adversarial attacks are limited. In Finlayson et al. (2019), adversarial examples are shown to deteriorate the diagnosis accuracy of deep learning based medical systems. These medical attack methods are mainly based on those from natural images (e.g., Fast Gradient Sign Method (FGSM) (Goodfellow et al., 2014) and Project Gradient Descent (PGD) (Madry et al., 2017), which are insufficiently developed for different types of medical data. As shown in Fig. 1, the adversarial examples generated by FGSM and PGD do not consistently decrease the network's performance in (b), (c), (f), (g), (j) and (k). The data variance in (a) and (e) leads to the inconsistent attack results by existing methods.

In this paper, we propose a medical image attack method to consistently produce adversarial perturbations that can fool deep medical diagnosis systems working with different medical data modalities. The perturbations are iteratively generated via taking partial derivatives of a well-defined objective function that is composed of a deviation loss term and a stabilized loss term with respect to the input. By maximizing the deviation loss term, the adversarial attack system enlarges the divergence between CNN predictions and the ground truth to have effective attack samples. To handle the aforementioned ubiquitous data noise issue in medical images, we propose a novel stabilization loss term as an extra regularization, which ensures a consistent deviation trajectory for the crafted attack samples. Meanwhile, the stabilization term avoids the local optima in the optimization process caused by the image noise.

The proposed stabilization loss term is designed to measure the difference between two CNN predictions, where the first prediction is from the crafted adversarial sample and the second one is from the same sample processed with a Gaussian smoothing. Given an adversarial example $A$ and its Gaussian smoothed result $\widetilde{A}$, the loss stabilization term constrains the corresponding CNN predictions (i.e., $f(A)$ and $f(\widetilde{A})$) to be similar via a minimization process. The intuition from the scale space optimization (Lindeberg, 1992) indicates that the minimization of $f(A)$ and $f(\widetilde{A})$ will exhaustively search the perturbation space to smooth the single spot for local optimum escape. We further analyze this stabilized loss term via KL-divergence and find that the CNN predictions are steered towards a fixed objective spot during iterations. This stabilization improves the attack effectiveness on different types of medical data including CT, fundus, and endoscopic images. We evaluate the proposed Stablized Medical Image Attack (SMIA) on several medical datasets (APT, 2019; EAD, 2019; Kag, 2015), including the recent COVID-19 (COV, 2019) lung CT. Thorough evaluations demonstrate that the proposed method is effective to produce perturbations that decrease the prediction accuracy of different medical diagnosis systems. Our investigation provides a guidance for strengthening the robustness of these medical systems towards adversarial attacks.

## 2 RELATED WORK

In this section, we literately review existing adversarial attack methods on both natural and medical images. Meanwhile, we survey relevant medical image analysis tasks where SMIA is deployed.

### 2.1 ADVERSARIAL ATTACK

There are extensive investigations on adversarial attacks for natural image classifications. In Goodfellow et al. (2014), FGSM was proposed to generate adversarial examples based on the CNN gradients. A DeepFool method was proposed in Moosavi-Dezfooli et al. (2016) to compute minimal perturbations based on classifier's linearization. In Moosavi-Dezfooli et al. (2017), an iterative algorithm was proposed to generate perturbations and showed the existence of a universal (image-agnostic) adversarial perturbations. In Baluja & Fischer (2017), a Transformation Network (ATN) was trained to generate adversarial examples without gradient involvement. The adversarial training and provable defense were proposed in Balunovic & Vechev (2020) to achieve both attack robustness and high accuracy. The capsule based reconstructive attack was proposed in Qin et al. (2020) to cause both misclassifications and reconstructive errors. Besides image classification, several attack methods were proposed for semantic segmentation, object detection and object tracking Jia et al. (2020). In Fischer et al. (2017); Dong et al. (2019), the classification based attacks were shown transferable to attack deep image segmentation results. The universal perturbations were demonstrated existing in Moosavi-Dezfooli et al. (2017). Moreover, a Dense Adversary Generation (DAG) method was proposed Xie et al. (2017) for both semantic segmentation and object detection attacks. The general idea of natural image attacks was to iteratively generate perturbations based on the CNN gradients to maximize the network predictions of adversarial examples and the ground truth labels. This idea was also reflected in Finlayson et al. (2019) to show the medical attacks. Different from existing methods, we propose a stabilized regularization term to ensure the consistent generation of adversarial perturbations, which are effective for different types of medical image datasets.

### 2.2 DEEP MEDICAL IMAGE ANALYSIS

The deep Convolutional Neural Networks (CNNs) have been shown effective to automatically analyze medical images (Litjens et al., 2017; Razzak et al., 2018). The common CADx applications include classifying the stage of disease, the detection and segmentation of organs and lesions.

**Disease classification.** Most medical systems formulate disease diagnosis as an image classification task. The types of diseases are predefined and each type corresponds to one category. During the classification process, there are single or multiple images as input for disease diagnosis. In Shen et al. (2015), a multi-scale CNN was proposed to capture the feature representation of lung nodule patches for accurate classification. A multi-instance layer and a multi-scale layer were proposed in Li et al. (2019) to diagnose diabetic macular edema and myopic macular degeneration. Besides diagnosing the types of disease, existing medical systems were also able to predict the disease status (Gulshan et al., 2016) by empirical status categorization.

**Organ and lesion detection.** The detection of organ and lesion is inspired by the object detection framework for natural images (e.g., Faster-RCNN (Ren et al., 2015), FPN (Lin et al., 2017), and Yolo (Redmon et al., 2016)). Besides, 3D spatial information of the medical data is explored in the 3D detection framework. In Ding et al. (2017), a 3D-CNN classification approach was proposed to classify lung nodule candidates that were previously detected by the Faster-RCNN detector. The RPN (Ren et al., 2015) was extended in Liao et al. (2019) to become 3D-RPN for 3D proposal generation to detect lung nodules. Different from the 3D detection framework, a multi-scale booster was proposed in Shao et al. (2019) with channel and spatial attentions that were integrated into FPN for suspicious lesion detection in 2D CT slices. The detection methods based on 2D image input reduced heavy computational cost brought in the 3D inputs for 3D detection methods.

**Organ and lesion segmentation.** The medical segmentation was significantly advanced via deep encoder-decoder structures (e.g., U-Net (Ronneberger et al., 2015)). This architecture contains a contracting path (i.e., encoder) to capture global context and a symmetric expanding path (i.e., decoder) to obtain precise localization. There were several works built upon U-Net. In Brosch et al. (2016), skip connections were utilized in the first and last convolutional layers to segment lesions in brain. A V-Net was proposed in Milletari et al. (2016) to segment brain's anatomical structures. It followed 3D U-Net structure consisting of 3D convolutional layers. A dual pathway and multi-scale 3D-CNN architecture was proposed in Kamnitsas et al. (2017) to generate both global and local lesion representations in the CNN for brain lesion segmentation. Existing medical methods mainly utilized deep encoder-decoder architectures for end-to-end segmentation of organs and lesions.

As illustrated above, deep medical diagnosis systems differ much from the CNN architectures that are developed for natural images. Moreover, the variance of different modalities of medical images is significantly larger than that of natural images. Therefore, the adversarial attacks designed for natural images are not often effective in the medical domain. Nevertheless, the limitations that are brought by huge network and data variance are effectively solved via our stabilized medical attack.

## 3 PROPOSED METHOD

In this section, we illustrate the details of medical image attacks. We first show the objective function of SMIA that consists of a loss deviation term and a loss stabilization term. The loss deviation term produces the perturbation to decrease the image analysis performance, while the loss stabilization term is consistently updated during iterations and constrains these perturbations to low variance. Then, we analyze how SMIA affects the generated perturbations during iterative optimization from the perspective of KL-divergence. The analysis is followed by a visualization showing the variance and cosine distance of perturbations by utilizing the loss stabilization term.

### 3.1 OBJECTIVE FUNCTION

As aforementioned, there are a loss deviation term (DEV) and a loss stabilization term (STA) in the objective function. The loss deviation term follows Goodfellow et al. (2014) to enlarge the difference between CNN predictions and the ground truth label. We denote the input image as $x$, model parameters as $\theta$, and the ground truth label of $x$ as $Y$. In the first iteration, the objective function of SMIA denoted as $\mathcal{L}_{\text{SMIA}}$ can be written as:

$$\mathcal{L}_{\text{SMIA}} = \mathcal{L}_{\text{DEV}} = \mathcal{L}(f(\theta, x), Y), \tag{1}$$

where $f(\theta, x)$ is the CNN predicted result of $x$ and $\mathcal{L}(\cdot)$ is the loss function. The perturbation $r$ can be computed by taking the partial derivatives of the objective function with respect to the input. It can be written as:

$$\begin{aligned} r &= \frac{\partial \mathcal{L}_{\text{SMIA}}}{\partial x} \\ &= \frac{\partial \mathcal{L}(f(\theta, x), Y)}{\partial x} \\ &= \nabla_x \mathcal{L}(f(\theta, x), Y). \end{aligned} \tag{2}$$

The perturbation is further refined to $\eta = \epsilon \cdot \text{sign}(r)$ where $\epsilon$ is a constant controlling the influence of $r$. We add $\eta$ to the input image $x$ as an adversarial example. The perturbation $\eta$ is iteratively learned

to enlarge the difference between $f(\theta, x)$ and $Y$. However, we observe that the objective function with the loss deviation term alone has a major unstable issue, especially for the medical images with large variances.

To tackle the instability problem, we introduce the loss stabilization term together with the loss deviation to form the new objective function, starting from the second iteration of the training process. We compute $\eta$ in the first iteration using Eq. 2. The loss stabilization term can be written as:

$$\mathcal{L}_{\mathrm{STA}} = -\mathcal{L}(f(\theta, x + \eta), f(\theta, x + W * \eta)), \tag{3}$$

where $W$ is a Gaussian kernel and convolves with the current perturbation $\eta$. This term enforces the CNN predictions of $x + \eta$ and $x + W * \eta$ similar under the current loss function. The objective function $\mathcal{L}_{\mathrm{SMIA}}$ we use during the following iterations can be written as:

$$
\begin{aligned}
\mathcal{L}_{\mathrm{SMIA}} &= \mathcal{L}_{\mathrm{DEV}} + \alpha \cdot \mathcal{L}_{\mathrm{STA}} \\
&= \mathcal{L}(f(\theta, x + \eta), Y) - \alpha \cdot \mathcal{L}(f(\theta, x + \eta), f(\theta, x + W * \eta)) \\
&= \mathcal{L}(f(\theta, \tilde{x}), Y) - \alpha \cdot \mathcal{L}(f(\theta, \tilde{x}), f(\theta, \tilde{x} + \eta')),
\end{aligned}
\tag{4}
$$

where $\alpha$ is the scalar balancing the influences of $\mathcal{L}_{\mathrm{DEV}}$ and $\mathcal{L}_{\mathrm{STA}}$, $\tilde{x} = x + \eta$ is the adversarial example sent to the CNN in the current iteration, and $\eta' = W * \eta - \eta$.

## 3.2 SMIA INTERPRETATION VIA KL-DIVERGENCE

The iterative generation of adversarial perturbation via SMIA is the process of maximizing the objective function shown in Eq. 4. In the following, we show how stabilization gradually arises and makes CNN prediction update to a constant value from the perspective of KL-divergence (Zhao et al., 2019; Shen et al., 2019). We elucidate our interpretation of SMIA via the image classification task where the loss function $\mathcal{L}$ is the cross entropy loss. The KL-divergence is utilized to measure the distribution difference between $f(\theta, \tilde{x})$ and $f(\theta, \tilde{x} + \eta')$, which are from the loss stabilization term as shown in Eq. 3.

We denote the input image as $x$ and there are $K$ categories in total. The ground truth label of $x$ is denoted as $Y = [y_1, y_2, ..., y_K]^T$ that is a one-hot label. The output of CNN $f(\theta, x)$ is a $K$ dimensional vector where each element represents the probability of $x$ belonging to the corresponding category. We denote the probability of $x$ belonging to $j$-th category ($j = 1, 2, ..., K$) as $p(y_j|x) = p_j(x)$. The KL-divergence $D_{\mathrm{KL}}$ measures the difference between $f(\theta, \tilde{x})$ and $f(\theta, \tilde{x} + \eta')$ and can be expanded via the second-order Taylor expansion (Shen et al., 2019) as follows:

$$
\begin{aligned}
D_{\mathrm{KL}}(f(\theta, \tilde{x}), f(\theta, \tilde{x} + \eta')) &= \mathbb{E}_y[\log \frac{p(y|\tilde{x})}{p(y|\tilde{x} + \eta')}] \\
&\approx \frac{1}{2}(\eta')^T \mathbf{G}_{\tilde{x}} \eta',
\end{aligned}
\tag{5}
$$

where $y$ is the random variable ranging from $y_1$ to $y_K$ under the distribution $f(\tilde{x})$, and $\mathbf{G}_{\tilde{x}} = \mathbb{E}_y[(\nabla_{\tilde{x}} \log p(y|\tilde{x}))(\nabla_{\tilde{x}} \log p(y|\tilde{x}))^T]$ is the Fisher Information Matrix of $\tilde{x}$. This matrix is designed to measure the variance of a distribution model.

During the perturbation generation process, SMIA iteratively minimizes the KL-divergence between $f(\theta, \tilde{x})$ and $f(\theta, \tilde{x} + \eta')$. As shown in Eq. 5, the minimization decreases the variances of $\eta'$ as well as seeking the smallest eigenvalue of $\mathbf{G}_{\tilde{x}}$. The variance of $\eta'$ is proportional to that of $\eta$, which indicates that the learned perturbations via SMIA are constrained to maintain low variance.

On the other hand, the eigenvalue computation of $\mathbf{G}_{\tilde{x}}$ is computationally intensive because the dimension of $\mathbf{G}_{\tilde{x}}$ is equal to the image size. We follow Zhao et al. (2019) to formulate $\mathbf{G}_{\tilde{x}}$ as a new matrix $\mathbf{G}_f$ through $\mathbf{G}_{\tilde{x}} = \mathbf{J}^T \mathbf{G}_f \mathbf{J}$. The term $\mathbf{G}_f$ is the Fisher Information Matrix of $f(\tilde{x})$ and the term $\mathbf{J}$ is the Jacobian matrix which can be computed as $\mathbf{J} = \frac{\partial f}{\partial \tilde{x}}$. As a result, the term $\mathbf{G}_f$ is a $K \times K$ positive definite matrix and formulated as:

$$\mathbf{G}_f = \mathbb{E}_y[(\nabla_f \log p(y|f))(\nabla_f \log p(y|f))^T]. \tag{6}$$

Therefore, the problem of minimizing eigenvalues of $\mathbf{G}_{\tilde{x}}$ is converted to the problem of minimizing the eigenvalues of $\mathbf{G}_f$. As the trace of a matrix equals the summation of its eigenvalues which are

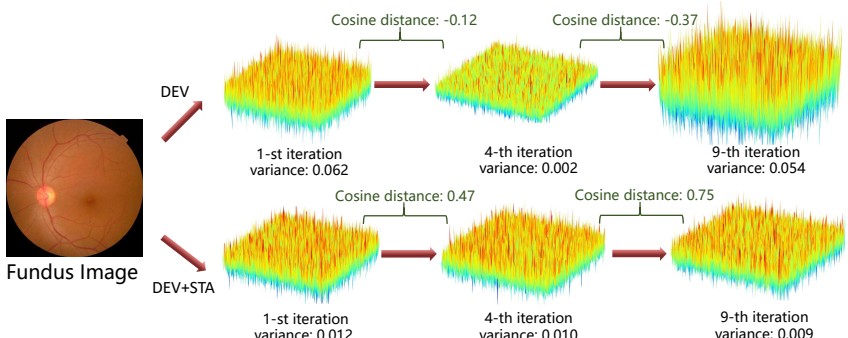

Figure 2: Visualizations of adversarial perturbations. We show how perturbation varies on the first, fourth, and ninth iterations, which is generated via $\mathcal{L}_{\text{DEV}}$ and $\mathcal{L}_{\text{DEV}} + \mathcal{L}_{\text{STA}}$ on an input fundus image. The perturbations produced via $\mathcal{L}_{\text{DEV}} + \mathcal{L}_{\text{STA}}$ have low variance and consistent cosine distance values, which accords with the analysis of the loss stabilization term via KL-divergence.

all positive, minimizing eigenvalues is equivalent to finding the smallest trace. The trace of $\mathbf{G}_f$ can be computed as:

$$
\begin{aligned}
tr(\mathbf{G}_f) &= tr(\mathbb{E}_y[(\nabla_f \log p(y|f))(\nabla_f \log p(y|f))^T]) \\
&= \int_y p(y|f)[tr(\nabla_f \log p(y|f))^T(\nabla_f \log p(y|f))] \\
&= \sum_{i=1}^{K} p_i \sum_{j=1}^{K} (\nabla_{p_j} \log p_i)^2 \\
&= \sum_{i=1}^{K} \frac{1}{p_i}.
\end{aligned}
\tag{7}
$$

Therefore, the objective function shown in Eq. 4 can be formulated as follows:

$$
\mathcal{L}_{\text{SMIA}} \propto \mathcal{L}(f(\theta, \tilde{x}), Y) - \alpha \cdot [\sum_{i=1}^{K} \frac{1}{p_i}] \cdot (\eta')^T \cdot \eta'.
\tag{8}
$$

When we iteratively add $\frac{\partial \mathcal{L}_{\text{SMIA}}}{\partial \tilde{x}}$ to the last adversarial example $\tilde{x}$, we are maximizing the values of $\mathcal{L}_{\text{SMIA}}$. Specifically, we maximize the first term while minimizing the second term in Eq. 8. The maximization of the first term resembles existing adversarial attack methods (e.g., FGSM and PGD). However, the minimization of the second term consists of minimizing both $\eta'$ and $\sum_{i=1}^{K} \frac{1}{p_i}$. When we minimize $\eta'$, we reduce the variance of the generated perturbation $\eta$. On the other hand, the optimal solution to $\min \sum_{i=1}^{K} \frac{1}{p_i}, \text{s. t.} \sum_{i=1}^{K} p_i = 1$ is $p_1 = p_2 = ... = p_K = \frac{1}{K}$. This indicates that the stabilization loss term enforces the CNN prediction to update towards a constant vector $[\frac{1}{K}, \frac{1}{K}, ..., \frac{1}{K}]$. To this end, the perturbations are learned to move consistently towards a fixed objective spot during different iterations in the CNN attack space. As this spot is not related to network structure nor data type, our loss stabilization term improves perturbation robustness that overcomes huge variations brought by different networks and different types of input data.

## 3.3 VISUALIZATIONS

The analysis on $\mathcal{L}_{\text{STA}}$ from KL-divergence indicates that the iteratively generated perturbations are in small variance and updated in a more consistent direction. We visualize the perturbations in different iterations in Fig. 2 to see how perturbations vary in practice. We send an input fundus image to the ResNet-50 (He et al., 2016) integrated with a graph convolutional network (Kipf & Welling, 2016) framework for image classification. The perturbations are generated via Eq. 1 and Eq. 4 independently.

Table 1: Ablation study on lung segmentation, artefact detection, and diabetic retinopathy grading, STA improves the attack performance while updating perturbations coherently.

| Methods | COVID-19 | | | EAD-2019 | | | APTOS-2019 | |
|---|---|---|---|---|---|---|---|---|
| | Acc(%)↓ | Jaccard(%)↓ | Cos(%)↑ | IOU(%)↓ | mAP (%)↓ | Cos(%)↑ | Acc(%)↓ | Cos(%)↑ |
| Clean | 87.82 | 93.57 | - | 22.86 | 43.36 | - | 81.82 | - |
| DEV | 69.16 | 83.66 | 60.00 | 6.82 | 17.28 | 64.98 | 69.35 | 90.34 |
| DEV + STA | **60.82** | **79.18** | **63.21** | **4.8** | **3.47** | **68.54** | **57.37** | **95.46** |

Fig. 2 shows the visual comparison of perturbations generated on the first, fourth, and ninth iterations. We denote the perturbations generated via $\mathcal{L}_{\mathrm{DEV}}$ (i.e., Eq. 1) as $P_{\mathrm{DEV}}$ and that generated via $\mathcal{L}_{\mathrm{DEV}} + \mathcal{L}_{\mathrm{STA}}$ (i.e., Eq. 4) as $P_{\mathrm{DEV+STA}}$. The visualization indicates that at the same iteration, the variance of $P_{\mathrm{DEV}}$ is larger than that of $P_{\mathrm{DEV+STA}}$. Meanwhile, during different iterations, there are severe variance oscillations in $P_{\mathrm{DEV}}$ while $P_{\mathrm{DEV+STA}}$ is consistently decreased. On the other side, we calculate the cosine distance between $P_{\mathrm{DEV}}$ and $P_{\mathrm{DEV+STA}}$ separately. The cosine distance indicates that $P_{\mathrm{DEV}}$ is updated in diverse directions while $P_{\mathrm{DEV+STA}}$ is more consistent. The visualization accords with the KL-divergence analysis that the proposed loss stabilization term decreases the perturbation variance and constrain a stable update direction towards a fixed objective spot in the CNN feature space.

## 4 EXPERIMENTS

We evaluate the proposed method on three medical image analysis tasks including diabetic retinopathy grading, artefact detection, and lung segmentation. The diabetic retinopathy grading is to classify fundus images into predefined categories for diabetes status estimation. The artefact detection is to detect specific artefacts like pixel saturations, motion blur, and specular reflections in the endoscopic images. Lung segmentation is to segment lung region from the whole CT slice. The medical data in one task is significantly different from that in others.

We use two datasets for diabetic retinopathy grading. One is the APTOS-2019 (APT, 2019) dataset with 3,662 fundus images. The other is a large-scale Kaggle-DR (Kag, 2015) dataset where we randomly select 11,000 fundus images from its original training set. Both APTOS-2019 and Kaggle-DR contains five defined categories. For artefact detection we use EAD-2019 (EAD, 2019) dataset with 2,500 images collected from endoscopic video frames and annotated artefact regions with seven defined categories. These detection images focus on multiple image modalities (i.e., gastroscopy, cystoscopy, gastro-oesophageal and colonoscopy), and are captured in multi-resolution with multi-modal (i.e., white light, fluorescence, and narrow band imaging). For lung segmentation, we use the COVID-19 dataset (COV, 2019) where there are 20 CT scans for lungs infected by COVID-19.

In practice, we use ResNet-50 (He et al., 2016) integrated with graph convolutional network (Kipf & Welling, 2016) to classify fundus images, the multi-scale booster framework (Shao et al., 2019) to detect artefact, and U-Net (Ronneberger et al., 2015) to segment infected lung regions. The proposed method is compared to other adversarial attack methods including FGSM (Goodfellow et al., 2014), PGD (Madry et al., 2017), DeepFool (Moosavi-Dezfooli et al., 2016), and DAG (Xie et al., 2017). The comparisons are made by adding these generated perturbations on the input images for decreased performance observation. For evaluation metrics, we use mean accuracy for medical image classification, IoU and mean Average Precision (mAP) for detection, and mean accuracy and Jaccard index for segmentation. During perturbation generation, we set $\epsilon$ as 0.05, 0.01, $5 \times 10^{-5}$ for lung segmentation, artefact detection and diabetic retinopathy grading respectively. We stop at the 10-th iteration for all the attack methods. The $\alpha$ in Eq. 4 is set as 1. We provide more results including using different Gaussian kernels $W$ and show the pseudo code in the supplementary files. Our implementation will be made available to the public.

### 4.1 ABLATION STUDY

The proposed SMIA introduces a stabilization loss term to contribute to the consistent perturbation generation process across medical image modalities and tasks. We evaluate how this stabilization term improves the attack performance of the deviation loss. Besides using the aforementioned metrics on three datasets, we measure the cosine distance between two consecutive perturbations to

Table 2: Attack performance on the EAD-2019 dataset with different iteration numbers.

| Iteration# | 15 | | 20 | | 25 | |
|---|---|---|---|---|---|---|
| | IOU(%) | mAP(%) | IOU(%) | mAP(%) | IOU(%) | mAP(%) |
| Clean | 22.86 | 43.36 | 22.86 | 43.36 | 22.86 | 43.36 |
| FGSM | 5.46 | 14.64 | 5.19 | 13.94 | 4.97 | 13.03 |
| PGD | 4.79 | 13.40 | 4.11 | 11.90 | 3.94 | 10.42 |
| DAG | 3.51 | 1.02 | 2.28 | 0.03 | 1.45 | 0.01 |
| SMIA | **1.02** | **0.24** | **0.50** | **0.02** | **0.11** | **0.01** |

Table 3: Attack performance on the EAD-2019 dataset with different $\epsilon$.

| $\epsilon$ | 0.005 | | 0.01 | | 0.015 | |
|---|---|---|---|---|---|---|
| | IOU(%) | mAP(%) | IOU(%) | mAP(%) | IOU(%) | mAP(%) |
| Clean | 22.86 | 43.36 | 22.86 | 43.36 | 22.86 | 43.36 |
| FGSM | 12.03 | 26.65 | 6.82 | 17.28 | 5.27 | 13.52 |
| PGD | 11.49 | 22.89 | 6.48 | 16.36 | 3.49 | 10.73 |
| DAG | **11.28** | **14.94** | 7.62 | **2.71** | 2.86 | **1.44** |
| SMIA | 11.34 | 15.92 | **4.8** | 3.47 | **1.18** | 3.25 |

calculate the similarity of their directions in the CNN attack space. We also calculate the percentage of perturbations with a positive cosine distance since a positive value indicates that the directions of perturbation update are consistent during the iterative generation process. After evaluating the STA term on three datasets, we analyze the influence of hyper-parameters to the attack performance on the EAD-2019 dataset across multiple attack methods.

Table 4: Attack performance on the EAD-2019 dataset with different $\alpha$ in Eq. 4.

| $\alpha$ | IOU(%) | mAP(%) |
|---|---|---|
| Clean | 22.86 | 43.36 |
| 0 | 6.82 | 17.28 |
| 0.75 | 5.94 | 12.18 |
| 1 | **4.80** | **3.47** |
| 1.25 | 4.93 | 4.05 |

Table 5: Numerical evaluations on the MNIST dataset.

| Methods | Accuracy(%) | Decrease(%) |
|---|---|---|
| Clean | 99.39 | - |
| FGSM | 75.90 | 23.63 |
| PGD | 74.88 | 24.66 |
| SMIA | 73.72 | 25.83 |

Table 1 shows the ablation study results on three datasets. We denote "Clean" as the results generated by medical diagnosis systems without adding perturbations. These results indicate the upper limit of these systems. In general, the perturbations generated via deviation loss term (DEV) decrease the recognition performance (i.e., segmentation, detection, and classification results in COVID-19, EAD-2019, and ATPOS-2019, respectively). By integrating loss stabilization term (STA), we are able to generate perturbations that decrease the recognition performance even further. On the other hand, the percentage of positive cosine distance values of DEV+STA is higher than DEV, which indicates the perturbations generated via DEV+STA are more consistent with each other during their iterative generation process. The results shown in this table indicate that STA improves the adversarial attack with DEV alone on both performance deterioration and perturbation consistency on three medical analysis scenarios.

Next, we analyze how the hyper-parameters of existing attack methods affect the attack performance on the EAD-2019 dataset. Tables 2-4 show the analysis results. We evaluate attack performance by using different iteration numbers in Table 2, different values of $\epsilon$ in Table 3, and different values of $\alpha$ in Table 4, respectively. The analysis on iteration numbers indicates that existing attack methods decrease the artefact detection performance by increasing the iteration numbers. SMIA significantly decreases the performance by using a limited number of iterations. The analysis on $\epsilon$ shows that SMIA performs favorably against existing methods when it is set as 0.01. Besides, SMIA decreases most when $\alpha = 1$. The hyper-parameter analysis shows our optimal parameter selections.

## 4.2 COMPARISONS WITH STATE-OF-THE-ART

We compare the proposed SMIA method with existing adversarial attack methods in the same medical analysis scenarios. Besides the prevalent FGSM and PGD methods, we involve DAG which is

Table 6: Comparison with state-of-the-art on lung segmentation, artefact detection, and diabetic retinopathy grading. The proposed SMIA method decreases recognition accuracies while producing perturbations with low variance.

| | COVID-19 | | | EAD-2019 | | | APTOS-2019 | | Kaggle-DR | | |
|---|---|---|---|---|---|---|---|---|---|---|---|
| | Acc(%) | Jac(%) | Var | IOU(%) | mAP(%) | Var | Acc(%) | Var | Acc(%) | Var | FPR(%) |
| Clean | 87.82 | 93.57 | - | 22.86 | 43.36 | - | 81.28 | - | 75.64 | - | 18.18 |
| FGSM | 69.16 | 83.66 | 0.087 | 6.82 | 17.28 | 0.062 | 69.35 | 0.052 | 58.91 | 0.064 | 25.56 |
| PGD | 69.40 | 83.69 | 0.081 | 6.48 | 16.36 | 0.086 | 67.46 | 0.126 | 56.89 | 0.085 | 26.09 |
| DeepFool | - | - | - | - | - | - | 68.23 | 0.073 | 56.57 | 0.069 | 27.01 |
| DAG | 68.96 | 90.87 | 0.054 | 7.62 | **2.71** | 0.059 | - | - | - | - | - |
| SMIA | **60.82** | **79.18** | **0.001** | **4.8** | 3.47 | **0.019** | **57.37** | **0.023** | **51.26** | **0.041** | **27.80** |

proposed for detection and segmentation of natural images, and DeepFool that is for disease image classification. All these methods are deployed on the medical diagnosis systems to show the recognition performance decrease. Meanwhile, we compute the variance of all the generated perturbations, which is shown as "Var" in the results.

Table 6 shows the evaluation performance. Under lung segmentation (i.e., COVID-19) and diabetic retinopathy grading (i.e., APTOS-2019 and Kaggle-DR) scenarios, our SMIA consistently decreases the recognition performance of medical systems. The drops in accuracy and Jaccard index brought by our method are more than those from existing methods. Specifically, the accuracy drop on the COVID-19 dataset brought by our method is around 9% more than the second best method (i.e., DAG). Moreover, the accuracy drop by using our method on the APTOS-19 dataset is around 10% more than the second best method (i.e., PGD). The false positive rate brought by our method is higher than other methods in Kaggle-DR. These performance drops indicate the effectiveness of our method to handle medical image data. Under the artefact detection scenario in EAD-2019, we observe that the decrease of mAP from DAG is more than ours. This is because DAG synthesizes pseudo labels for each proposal from the detection network to specifically attack the classifier of the detector. Nevertheless, its performance decrease is not as significant as ours (i.e., 7.62% v.s. 4.8%) under the IoU metric that measures the bounding box overlap ratios. On the other hand, the perturbation variance (i.e., "Var") of our method is lower than other attack methods on all the evaluation datasets. The decreased performance and smaller perturbations indicate the effectiveness of our SMIA method for attacking medical diagnosis systems.

Besides medical image datasets, we evaluate SMIA on a hand-written image dataset MNIST (Deng, 2012) with existing methods. Table 5 shows the results. Our SMIA method decreases the recognition accuracy more than that of FGSM and PGD. This indicates that SMIA is effective to attack not only medical images, but also images from other fields.

## 5 CONCLUSION

We proposed a medical image adversarial attack method to diagnose current medical systems. The proposed method iteratively generates adversarial perturbations by maximizing the deviation loss term and minimizing the loss stabilization term. For an adversarial example at the current iteration, the difference between its CNN prediction and the corresponding ground truth label is enlarged, while the CNN predictions of its smoothed input and itself are similar. The similar predictions constrain the perturbations to be generated in a relatively flat region in the CNN feature space. During consecutive iterations, the perturbations are updated from one flat region to another. Thus the proposed method can search the perturbation space to smooth the single spot for local optimum escape. Compared to the perturbation movement among single spots that are brought by only using loss deviation term, the loss stabilization term improves the attack performance by regularizing the perturbation movement to be stable. Further analysis with the KL-divergence shows that the minimization of loss stabilization term is to regularize the perturbations to move towards a fixed objective spot while their variances are reduced. Both the visualization and experimental results have shown the effectiveness of our attack method to figure out the limitations of medical diagnosis systems for further improvement.

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
