# OpenReview forum: "Stabilized Medical Image Attacks"
_ICLR.cc/2021/Conference — ICLR 2021 Spotlight_

### Official Review · AnonReviewer1 · 2020-10-27
**A stabilized adversarial attack method for various types of medical data.**

**Rating:** 8
**Confidence:** 5

**Review:**

The authors present a universal medical attack method that can consistently produce adversarial examples across several medical imaging domains. The authors achieve this by developing a novel objective function that includes two terms, which they refer to as stabilized medical attack (SMA). The first term is the loss deviation term, inspired by the conventional fast gradient sign method, which enlarges the difference between CNN predictions and ground truth labels. The second term, a regularizer, is the loss stabilization term that enforces consistent predictions between the adversarial image and the Gaussian smoothed version of the adversarial image. The authors then provide an insightful interpretation of their SMA loss via KL divergence. The derivation demonstrates that perturbations consistently move towards a fixed location in the SAM objective landscape during successive iterations of gradient ascent. This method increases perturbation robustness by overcoming huge variations that result from different types of medical imaging data. The authors provide an illustrative figure (Fig.2) to demonstrate that both the variance and direction of the adversarial perturbation remain stable and consistent across multiple iterations, compared to using the deviation loss alone. The authors then perform an ablation study to demonstrate the DEV + STA loss results in a significantly greater reduction in model performance across medical imaging datasets compared to DEV loss alone. Finally, compared to the state-of-the-art adversarial methods, SMA results in the greatest reduction in performance for all datasets.

Pros:
This is an excellent paper. Succinct and clear, addressing a known problem with any CAD system. They provide good justification of the technique. They provide sufficient mathematical detail to follow the KL divergence derivation. Both empirical studies are sound and demonstrate the expected results. Additional examples are provided in the supplementary materials. The methods are correct. The empirical methodology is correct. The paper is generally clear.

Cons:
It would be helpful for the reader/audience to have an associated figure that graphically demonstrated the point made in 3.2. There is a compelling geometric intuition behind the idea expressed in the text above Sec 3.3 that would help present the contribution of adding the STA term.

---

### Official Review · AnonReviewer4 · 2020-10-28
**Title needs to be more specific**

**Rating:** 7
**Confidence:** 4

**Review:**

The authors proposed to introduce a combination of a loss deviation term and a loss stabilization term to generate more consistent adversarial perturbations on medical images. The loss deviation term increases the divergence between the CNN prediction of an adversarial example and its ground truth label. At the same time, the loss stabilization term ensures similar CNN predictions of this example and its smoothed input. The authors tested against 3 different medical image datasets obtained by different modalities. The proposed strategy seems straightforward and the benefits are clearly demonstrated with these three datasets.

Overall, I vote for good paper, accept, after minor revise. The proposed strategy seems straightforward and the benefits are clearly demonstrated with these three datasets. This approach may contribute to further improve CNN-based medical image diagnosis systems.

Minor points:

Since the authors demonstrated only with medical image dataset, the title Stabilized “Medical Attacks” seems misleading and inaccurate. There are many other data types in medical field such as text, wave form, numerical values, etc, thus the title should be more specific, at least “Medical Image Attacks”. Also the term “universal” is used several times in the main text, and it has the same issue since the authors tested only three medical image datasets while there are many more different modalities exist in the real medical field even if limit the data type to an image.
Conversely, the strategy, introducing a stabilization loss, seems not quite specific to the characteristics of medical image, rather potentially more general. Thus it is quite curious to see how this strategy can be generalized toward image datasets in other fields.

---

### Official Review · AnonReviewer3 · 2020-10-29
**proposes to use a regularization term for stabilizing the perturbation trajectories in generating adversarial examples for medical image tasks**

**Rating:** 7
**Confidence:** 3

**Review:**

The paper proposes to use a regularization term for stabilizing the perturbation trajectories in generating adversarial examples for medical image tasks. More specifically, they introduce a loss stabilization term which forces perturbed inputs to be close to smoothed perturbed inputs in the CNN output space. The authors give a shallow analysis which demonstrates that this regularizer forces convergence of softmax output to a uniform distribution (i.e. the maximum entropy distribution). In addition, their theoretical analysis yields a practical implementation for their loss term. The authors provide one visualization to demonstrate the differences in variance of the perturbation and variance in the directions of perturbation. Finally, they demonstrate the effectiveness of their method on three medical image datasets for separate computer vision tasks by comparing with the state of the art methods for adversarial attacks. I like the idea of using regularization in the generation of adversarial examples. In particular, the theoretical motivation to control the perturbation variance and output distribution is thought provoking. I believe more generalization and deeper analysis would be beneficial to future deep learning research, and for this primary reason do I vote to accept. My primary concern for the paper is the clarity of the presentation, which I explain in the cons section.
Pros:

1. The paper presents a theoretically founded, and (somewhat) experimentally validated application of regularization for generating adversarial examples for multi-modal tasks and data for medical images.

2. The proposed method is simple and elegant. It is theoretically well-founded and easily implemented.

3. The paper provides good initial results, showing that to some degree their method is generalizable. The ablation study provides a good basis for demonstrating the effectiveness of the proposed regularization.
Cons:

1. The datasets used are rather small and not well known within medical imaging community. I suggest expanding this study to include larger datasets (10,000+ images) and include other well-founded datasets for medical tasks (see literature cited by the paper). This would demonstrate greater generalization of the proposed method, especially in the case that this method generalizes to large, varied datasets.

2. Because the real danger of adversarial attacks in medical images concerns false positives, the generation of adversarial examples that yield false positives would be helpful. Experiments regarding this would be even better.

3. Could include visuals for multiple datasets and multiple tasks.

4. Should include information on how the hyper-parameters were determined and the results of such validation. Particularly, epsilon, alpha, and stopping iteration. In addition, why was the second iteration chosen to introduce the regularization?

5. On page 4, the authors claim “the limitations that are brought by huge network and data variance are effectively solved via our stabilized medical attack”. Why is this the case? While the method does reduce the variance in the perturbation and the direction of the perturbations, can you explain in the paper why this is inherent to deep networks and a variety of data modalities? The introduction could also touch on previous work demonstrating the difficulties adversarial examples have for differing data modalities (i.e. why, as is claimed, do previous methods not generalize away from natural images?)

---

### Decision · Program_Chairs · 2021-01-07
**Final Decision**

**Decision:**

Accept (Spotlight)

**Comment:**

 The paper proposes to use a regularization term for stabilizing the perturbation trajectories in generating adversarial examples for medical image tasks. The authors tested the effectiveness of their proposal on different medical image datasets obtained by different modalities, and the experimental results are generally encouraging.
All the reviewers see the value of the paper and give positive comments. At the same time, they also point out some aspects for further improvement, including
1)	The datasets used are relatively small
2)	The title is a little misleading since the paper only tackles the image attacks (but the title is stabilized medical attacks).
3)	Case studies and visualization are needed to help people better understand the paper

The authors have done a good job in their rebuttal and paper revision, by adding experiments on larger datasets, changing the title to “stabilized medical image attacks”, and adding some geometric figures for better illustration. These have largely addressed the concerns of the reviewers, and we see no problem with accepting the paper.